# Nuts: Natural Pleiotropic Nutraceuticals

**DOI:** 10.3390/nu13093269

**Published:** 2021-09-19

**Authors:** Emilio Ros, Annapoorna Singh, James H. O’Keefe

**Affiliations:** 1Lipid Clinic, Endocrinology and Nutrition Service, Institut d’Investigacions Biomediques August Pi Sunyer, Hospital Clinic, University of Barcelona, 08036 Barcelona, Spain; 2CIBER Fisiopatología de la Obesidad y Nutrición (CIBEROBN), Instituto de Salud Carlos III, 28029 Madrid, Spain; 3Saint Luke’s Mid America Heart Institute, University of Missouri-Kansas City, Kansas City, MO 64110, USA; singhann@umkc.edu (A.S.); jokeefe@saintlukeskc.org (J.H.O.)

**Keywords:** tree nuts, peanuts, fatty acids, prospective studies, randomized clinical trials, cardiovascular risk, type-2 diabetes, cancer, hypertension, cognitive function, mortality, body weight, blood lipids, inflammation, PREDIMED

## Abstract

Common nuts (tree nuts and peanuts) are energy-dense foods that nature has gifted with a complex matrix of beneficial nutrients and bioactives, including monounsaturated and polyunsaturated fatty acids, high-quality protein, fiber, non-sodium minerals, tocopherols, phytosterols, and antioxidant phenolics. These nut components synergize to favorably influence metabolic and vascular physiology pathways, ameliorate cardiovascular risk factors and improve cardiovascular prognosis. There is increasing evidence that nuts positively impact myriad other health outcomes as well. Nut consumption is correlated with lower cancer incidence and cancer mortality, and decreased all-cause mortality. Favorable effects on cognitive function and depression have also been reported. Randomized controlled trials consistently show nuts have a cholesterol-lowering effect. Nut consumption also confers modest improvements on glycemic control, blood pressure (BP), endothelial function, and inflammation. Although nuts are energy-dense foods, they do not predispose to obesity, and in fact may even help in weight loss. Tree nuts and peanuts, but not peanut butter, generally produce similar positive effects on outcomes. First level evidence from the PREDIMED trial shows that, in the context of a Mediterranean diet, consumption of 30 g/d of nuts (walnuts, almonds, and hazelnuts) significantly lowered the risk of a composite endpoint of major adverse cardiovascular events (myocardial infarction, stroke, and death from cardiovascular disease) by ≈30% after intervention for 5 y. Impressively, the nut-supplemented diet reduced stroke risk by 45%. As they are rich in salutary bioactive compounds and beneficially impact various health outcomes, nuts can be considered natural pleiotropic nutraceuticals.

## 1. Introduction

Common tree nuts include almonds, Brazil nuts, cashews, hazelnuts, macadamias, pecans, pine nuts, pistachios and walnuts. Botanically, the peanut (*Arachis Hypogaea*) is a legume, but it has a nutrient profile that is similar to the tree nuts listed above, which qualifies peanuts to be included in the nut food group [1]. The impact of nut consumption on health outcomes has been extensively investigated since the publication in 1992 of the pioneering Adventist Health Study, in which nut consumption was associated for the first time with a lower risk of coronary heart disease (CHD) [2]. Soon after, a landmark randomized clinical trial (RCT) demonstrated that walnut consumption significantly lowered blood cholesterol [3].

Nuts are nutrient-rich foods that have been a staple of humankind’s diet throughout our long evolutionary history [4]. However, during the last century, most people in industrialized nations have markedly reduced their consumption of nuts, so that now nuts comprise only a marginal source of dietary energy, except for vegetarians, health-conscious groups such as Seventh Day Adventists, and individuals following diets based on whole natural foods [5]. Tellingly, in the last two decades nut consumption has increased in Western countries in parallel with the United States (US) Food and Drug Administration’s issue of a health claim that nut consumption is associated with a reduced risk of CHD [6], inclusion of nuts in many guidelines for health promotion [7], and wide media advertising of their beneficial health effects.

The scientific evidence behind nuts as health-promoting foods stems from both abundant epidemiological observations suggesting that their regular consumption relates inversely to incidence of and mortality from major non-communicable diseases [8] and from RCTs disclosing a consistent cholesterol-lowering effect of nut diets [9]. The mechanisms for these salutary effects include the optimal nutrient composition of nuts, their satiating effects, and their tendency to displace other less healthy foods.

Contrary to the popular belief that, due to the high energy content of nuts, their consumption has a fattening effect, evidence from both epidemiological studies and RCTs suggests that their regular consumption does not lead to increased body weight and may even promote weight loss [10]. This review summarizes current knowledge on the increasingly important topic of nuts as health-promoting foods and their sizable contribution to the nutritional quality of the diet, while laying out the scientific basis to consider them as natural pleiotropic nutraceuticals.

## 2. Data Sources and Selection of Studies on Nuts and Human Health

For this narrative review we conducted a comprehensive search of the PubMed^®^/MEDLINE^®^ (https://www.ncbi.nlm.nih.gov/pubmed/ (accessed up to 31 July 2021)) database through July 2021 for English language articles of epidemiological and clinical studies illustrating the effects of exposure to nuts (tree nuts and peanuts) and their components (mainly peanut butter) on health outcomes, and the latest reviews and meta-analyses of these studies. Meta-analyses pooling data from nuts and seeds were excluded. We also searched the references from original research studies and reviews, as well as articles citing clinical studies, reviews, and meta-analyses, as listed by the publishers of individual articles in their websites. Given that the information of the various meta-analyses on the same outcome tends to be redundant, each successive one synthesizing the results of the same studies plus newly published ones, for each outcome, only the information from the most recent meta-analysis is discussed. However, older systematic reviews may be cited if they contain relevant information (i.e., dose–response analyses) not covered in the subsequent meta-analyses. For completeness, the data from well-designed cohort studies published after each specific meta-analysis are also reviewed.

Data were examined for relevance, quality, consistency and independently extracted by the two senior authors (ER, JHO), who reached an agreement when in doubt about a specific citation. Given that few RCTs on the effects of nut consumption on clinical end points are available, we obtain the core of scientific evidence from epidemiologic studies relating frequency of nut consumption to disease outcomes and RCTs of nut-enriched versus control diets for effects on intermediate end points, with particular attention to meta-analyses of such studies. 

## 3. Historical Aspects

Archeological sites throughout the world have produced proof of consumption of hard-shelled nuts by ancient humans going back to the mid-Pleistocene, one million years ago. The oldest evidence of cultivation of the common tree nuts almonds (*Prunus amigdalis*), hazelnuts (*Corylus avellana*), walnuts (English walnuts, *Juglans regia*), and pistachios (*Pistachia vera*) is from Asia, spanning from China to the Middle-East and the Anatolian peninsula (modern Turkey). These trees were subsequently cultivated in Greece, then in the territories of the Roman Empire and the Iberian peninsula, and were extended to all of Europe during the Middle Ages. In the 16th century, in one of the first and boldest food globalizations, European colonizers introduced these tree nuts to the Americas and brought those native to the Americas back to Europe [11]. In North America, there were native hazelnuts and walnuts—the so-called black walnut (*Juglans nigra*), as well as other indigenous nuts such as pecans (*Carya illinoinensis*), while cashews (*Anacardium occidentale*) and Brazil nuts (*Bertholettia excelsa*) are native to South America. Another popular tree nut, macadamias (*Macadamia integrifolia*), is native to Australia. The common pine nuts (*Pinus pinea*) are often obtained from natural forests, mostly in the Mediterranean region, but they are also native to China and North America.

Peanuts (also called groundnuts in some areas) were first cultivated from wild varieties by the ancient Incas from Peru. European explorers during the 16th century first discovered peanut plants, which were being cultivated in Brazil and Mexico, and transported peanuts back to Spain. From there, traders and explorers exported peanuts to Asia and Africa, and eventually to North America in the 1700s. Peanut butter was developed more than a century ago as a soft protein meal for people with poor dentition [12]. Today, peanuts and peanut butter are popular; Americans per capita eat about 6 pounds per year.

In Europe, nut supply is highest in Mediterranean countries [13]. Indeed, nuts are an integral component and a defining feature of the traditional Mediterranean diet, a dietary pattern characterized by high consumption of vegetables, fruits, nuts, olive oil, cereal grains; moderate consumption of fish and alcohol—mostly wine; and a low consumption of dairy products, red meat and meat products, and sweets [14]. Nuts can be are incorporated into the usual diet in different ways, as snacks, mixed in meals or desserts and may be eaten whole (fresh or roasted), in spreads (i.e., almond paste, peanut butter), as oils or hidden in commercial products, sauces, pastries, cakes, ice creams, and baked goods. Due to their high energy content and tasty nature, nuts have been widely introduced into sports snacks and supplements.

## 4. Nutrient Content

Nuts are nutrient dense foods, coming only after vegetable oils as the natural plant food richest in fat. Their total fat content as percent of weight ranges from 44% in cashews to 76% in macadamias, and they provide 23 to 30 kJ/g (Table 1) [15,16]. However, the fatty acid composition of nuts is salutary because they have a low saturated fatty acid (SFA) content (range, 4% to 16%) and nearly one-half of their total fat content is formed by unsaturated fat—specifically, monounsaturated fatty acids (MUFA) in most nuts, polyunsaturated fatty acids (PUFA) predominating over MUFA in pine nuts, similar amounts of MUFA and PUFA in Brazil nuts, and mostly PUFA in walnuts (Figure 1). Of note, with around 10 g per 100 g, walnuts are particularly rich in α-linolenic acid (ALA), the plant-derived essential omega-3 fatty acid [1]. The favorable lipid content of nuts is an important contributor to the beneficial health effects conferred by their frequent consumption.

Nuts are also contain other macronutrients and bioactives reputed as beneficial for health outcomes. They are a good source of vegetable protein (between 8% and 25% of energy) and are known to have a sizable content of the amino acid L-arginine, which is the substrate for the synthesis of endothelium-derived nitric oxide (NO), the main endogenous vasodilator and blood pressure (BP) regulator [17]. This explains in part why nut consumption helps improve endothelial function and may lower BP. Additionally, nuts are a good source of dietary fiber, ranging from 3 to 12.5 g per 100 g (Table 1) [15,16]. Indeed, a standard 1-oz (28-g) serving of nuts provides 5–10% of daily fiber requirements [1].

Among other nut components, there are several micronutrients that have salutary effects when taken in at doses beyond those necessary to prevent deficiency states. Nuts contain considerable amounts of the B-vitamin folate, peanuts being richest [15,18]. Almonds and hazelnuts are good sources of the antioxidant vitamins including tocopherols (e.g., vitamin E), while all nuts contain polyphenols, which are powerful antioxidants required to protect the germ from oxidative stress and preserve the reproductive potential of the seed [19]. Due to their protective characteristics, most polyphenols reside in the outer peel of nuts (between the shell and the nut), a good reason to eat raw, unpeeled nuts when possible. Walnuts, pistachios, and pecans have the highest polyphenol content (Table 2).

Nuts are devoid of cholesterol, but their fat fraction contains chemically related non-cholesterol sterols, which belong to a heterogeneous group of molecules known as plant sterols or phytosterols. These compounds are non-nutritive plant components that play a structural role in their cell membranes just as cholesterol does in animal cell membranes [20]. Phytosterols interfere with cholesterol absorption, thus helping lower blood cholesterol concentration when present in doses of 1 g or higher in the intestinal lumen. The mechanism of action of phytosterols depends on their hydrophobic nature, as they have a large hydrocarbon molecule with a higher affinity for micelles than has cholesterol. As phytosterols displace cholesterol from micelles, the amount of sterol available for absorption is reduced. Pistachios and almonds are highest in phytosterols (Table 1). Not unexpectedly, evidence has been provided that phytosterols contribute to the cholesterol-lowering effect of nut consumption [21].

Nuts are also a rich source of beneficial minerals, such as calcium, magnesium, and potassium (Table 2). As in most plant foods, the sodium content of nuts is very low. Low sodium intake coupled with high intake of calcium, magnesium and potassium is associated with protection against hypertension, insulin resistance, and cardiovascular (CV) disease (CVD) [22], besides counteracting bone demineralization. Even lightly salted nuts retain a relatively low sodium content.

In summary, the macronutrients, micronutrients, and phytochemicals of nuts have all been documented to contribute to beneficial health outcomes, particularly a reduced risk of CVD and related metabolic alterations. As shown in Figure 2, bioactive nut components synergize to affect multiple metabolic and vascular physiology pathways leading to decreased cardiometabolic risk. For these reasons, whole unprocessed nuts may be considered as natural pleiotropic nutraceuticals. As such, daily consumption of nuts should be considered an essential feature of a health-promoting diet.

## 5. Nut Consumption and Health Outcomes

The bulk of evidence concerning the effects of nuts on health outcomes stems from prospective studies, the results of which have been summarized in numerous systematic reviews and meta-analyses conducted over the last two decades. Many RCTs have also been conducted examining the effects of nuts on intermediate risk factors, and corresponding meta-analyses have been published. As only one seminal RCT, the PREDIMED study [23], has assessed the effects of a long-term nut-enriched diet on hard CVD outcomes, it will be discussed separately from other RCTs.

### 5.1. CVD Incidence and Mortality

CVD, mainly CHD and stroke, are the leading causes of death globally. Most CVD could be avoided by addressing and modifying behavioral risk factors, such as incorporating healthy dietary habits [24]. In the last three decades, considerable evidence has accumulated on the effects of frequent nut consumption on CVD outcomes. The most recent systematic review and meta-analysis of 19 prospective cohort studies by Becerra-Tomás et al. [25] found an inverse association between total nut consumption (comparing highest vs. lowest categories) and CVD incidence (Relative Risk [RR] = 0.85; 95% Confidence Interval [CI], 0.80, 0.91; 3 studies), CVD mortality (RR = 0.77; 95%CI, 0.72, 0.82; 14 studies), CHD incidence (RR = 0.82; 95% CI, 0.69, 0.96; 7 studies), CHD mortality (RR = 0.76; 95% CI, 0.67, 0.86; 11 studies), and stroke mortality (RR = 0.83; 95% CI, 0.75, 0.93; 11 studies). No association was ascertained with incident stroke, either ischemic or hemorrhagic, in seven and five studies, respectively. Regarding specific nut types, the reduced risk was noted with tree nuts and peanuts for most CVD outcomes, but not with peanut butter. However, concerning stroke mortality, reduced risk was found for high versus low consumption of peanuts (RR = 0.85; 95% CI, 0.79, 0.92), but not tree nuts. In dose–response analyses, total nut consumption and CVD outcomes showed non-linear inverse associations, with risk reductions up to a consumption of 5 g/day (stroke mortality), 10 g/day (CVD incidence), and 15–20 g/day (CVD and CHD mortality), namely, no further significant reductions were observed above these amounts. The findings of this meta-analysis concur with those of an earlier systematic review by Aune et al. [8].

In proof of the interest of the topic, data from additional large prospective cohort studies relating nut consumption to CVD outcomes, principally CVD mortality, have been released after that meta-analysis [25]. In an analysis of 16,217 men and women with diabetes from the prospective Nurses’ Health Study (NHS)-I and -II and the Health Professionals Follow-Up Study (HPFS), highest versus lowest total nut consumption was associated with a lower risk of CVD and CHD incidence and all-cause mortality, with RRs similar to those ascertained in the mentioned meta-analysis, but not with stroke incidence and mortality [26]. In these cohorts, only tree nuts, not peanuts, were associated with reduced CVD outcomes. In a very large (*n* = 566,398) population-based prospective study in the US with a median follow-up of 15.5 y, data on cause-specific mortality confirmed the inverse association between higher total nut consumption and CVD deaths (Hazard Ratio [HR], 0.70; 95% CI, 0.66, 0.74), while no association for peanut butter consumption was found [27]. The Prospective Urban and Rural Epidemiology (PURE) study, conducted in 16 countries from 5 continents, examined nut consumption in relation to CVD outcomes in 124,329 participants followed for a median of 9.5 y [28]. Overall, CV mortality was lower (RR = 0.72, 95% CI, 0.56, 0.92) in high nut consumers, but no significant effects were detected for CHD or stroke. In the Women’s Health Study (*n* = 39,167) with a mean follow-up of 19 y, higher versus lower nut consumption was associated with lower CVD mortality (HR = 0.73; 95% CI, 0.61, 0.87) [29]. In a recent Iranian population-based prospective cohort study comprised of 6504 participants, those in the highest quartile of nut consumption had a markedly decreased CVD risk (HR = 0.57, 95% CI, 0.47, 0.70) [30]. In a population-based prospective study from Japan (*n* = 31,552), even though participants consumed very low amounts of nuts (1.6 g/d on average), mostly peanuts, higher versus lower peanut consumption was associated with reduced CVD mortality in women, while only trends towards inverse associations were found in men for peanuts and in both sexes for total nuts [31].

Another report from the large NHS and HPFS prospective cohorts relates nut consumption to CVD risk in a unique way by estimating risk associated with changes in nut consumption, either increases or decreases, during 4-year periods [32]. The researchers found that increasing consumption of total nuts, tree nuts, walnuts and peanuts, but not peanut butter, is associated with reduced risk of total CVD, CHD and stroke, while the converse (increase in risk of CVD and stroke) occurred when participants decreased nut consumption. This is one of few epidemiologic studies supporting a positive effect of nuts on stroke risk, though the PREDIMED RCT showed a marked reduction in stroke in those assigned to the Mediterranean diet with nuts arm [23].

Finally, two very large population surveys in Europe [33] and Latin America [34] analyzed the contribution of dietary factors to CVD mortality and found that one of the most important factors, accounting for the largest number of cardiometabolic deaths, was low nut and seed consumption.

Few prospective studies have examined the relationship of nut consumption with two additional CVD outcomes: atrial fibrillation and heart failure. The cited meta-analysis [25] synthetized data from two cohorts that examined the association of highest vs. lowest nut consumption categories with atrial fibrillation and found a RR of 0.85 (95% CI, 0.73, 0.99). On the other hand, no effect of nuts on heart failure (two studies) was observed [25].

In summary, consistent data from numerous large, well-conducted prospective studies and meta-analyses suggest that nuts are potent cardioprotective foods. The effect of higher nut consumption is strongest on CVD and CHD mortality, with reductions of 25–30%, followed by CVD and CHD incidence and stroke mortality (15–18% reduction), while effects on stroke incidence are less consistent. Total nuts, tree nuts and peanuts, but not peanut butter, generally share the same positive effects on CVD risk. These effects are likely ascribable to nuts’ high content of healthy nutrients, such as PUFA, MUFA, non-sodium minerals, vitamins, and polyphenols and their potential to improve intermediate risk factors of CVD, as discussed in the corresponding section.

### 5.2. Hypertension Incidence and Mortality

The most recent review of epidemiological studies concerning total nut consumption in relation to cardiometabolic outcomes analyzes data from three meta-analyses of prospective studies with outcomes on incident hypertension [35]. An average 15% risk reduction, which was fairly constant in the three meta-analyses, was apparent when comparing high vs. low categories of total nut consumption. Based on data from four prospective studies with 11,962 incident hypertension cases, a 2017 dose–response meta-analysis (included in the review) estimated a 30% attenuation of hypertension risk for each daily serving (1-oz or 28 g) of nuts (RR = 0.70; 95% CI, 0.45, 1.08), with a linear dose–response [36]. No further prospective studies analyzing exposure to nuts in relation to hypertension risk have been published since that review.

In conclusion, in prospective studies nut consumption is associated with a consistent reduction of incident hypertension. As hypertension is the principal risk factor for stroke, this evidence clashes with the generally null epidemiological association of nut consumption with incident stroke, albeit, as discussed, increasing nut consumption is associated with lower stroke mortality. There is, however, sound RCT evidence that nut consumption lowers BP, as discussed in the sections on intermediate markers and health effects of nuts in the PREDIMED trial.

### 5.3. Diabetes Incidence and Mortality

The effects of nut consumption on risk of type-2 diabetes mellitus (T2D) in epidemiological studies have mostly been inconclusive and controversial [35,37]. A recent meta-analysis of nine studies (six prospective, three cross-sectional) published between 2002 and 2018 reported no association between extremes of consumption of total nuts, tree nuts or peanuts and the risk of T2D [38]. Walnuts, however, appeared to behave differently, as one large prospective study from the NHS and HPFS cohorts included in the meta-analysis found that walnut consumption related inversely to T2D risk (RR = 0.76; 95% CI, 0.62, 0.94) [39], while another large cross-sectional study found an even more beneficial effect of walnuts on T2D (RR = 0.47; 95% CI, 0.31, 0.71) [40].

Surprisingly, peanut butter, assessed in two early prospective studies, was inversely associated with T2D risk in the pooled estimate (RR = 0.87; 95% CI, 0.77, 0.98). Yet, only the results of one of the two analyzed studies favored peanut butter for T2D risk. Notably, risk estimates for total nuts changed substantially from nonsignificant to significant for lower T2D risk when time-updated measurement of body mass index (BMI) obtained during follow-up was excluded from the model (RR = 0.85; 95% CI, 0.75, 0.95), which supports body weight changes as a mediator of the reduction in T2D risk [38]. In fact, there is increasing evidence from prospective studies that regular nut consumption is associated with less long-term weight gain and a lower risk of obesity, while short-term RCTs confirm the lack of fattening effect of nuts [35]. Hence, given that long-term nut consumption is associated with a lower BMI, adjustment for BMI may conceal the true relationship between nuts and T2D, as also highlighted in the review of meta-analyses of nut studies by Kim et al. [35].

The main cause of death in T2D is CVD. Regarding nut consumption in relation to T2D mortality, the analysis of data from participants with T2D in the prospective cohorts of the NHS-I, NHS-II and HPFS showed that highest vs. lowest total nut consumption was associated with a 25% lower reduced CVD mortality and a 27% lower all-cause mortality [26]. In regard to T2D mortality, these data concur with the results of the earlier meta-analysis of Aune et al. [8].

In summary, epidemiological evidence suggests that consumption of total nuts is associated with a reduced T2D risk, which is mediated by nut-associated favorable weight changes that obscure the relationship when BMI is entered as covariate in adjustment models. RCTs also indicate favorable effects of nuts on glycemic control, as discussed in the section of intermediate markers below. Among the tree nuts, to date only the walnut has been associated with lower T2D risk. The unique nutrient composition of walnuts, rich in ALA, highly bioactive polyphenols and melatonin [41], may explain their differential effects on health outcomes, including T2D. A note of caution is necessary when considering the positive effect of peanut butter on T2D risk, as it was only ascertained in one prospective study. Additional large prospective studies are warranted to further elucidate the effects of nuts on T2D.

### 5.4. Cancer Incidence and Mortality

Cancer is a major cause of death and constitutes a huge public health hazard worldwide, as the global burden of cancer is expected to increase to 29.5 million new cancer cases and 16.4 million cancer-related deaths by 2040 [42]. At least 40% of cancers could be prevented by addressing modifiable risk factors such as tobacco use, dietary carcinogens, sedentary lifestyle, obesity and infectious agents [43]. In particular, adherence to a wholesome eating pattern such as a traditional Mediterranean diet has been shown to reduce the risk of some cancers by 4% to 57% [44]. There is substantial epidemiological evidence suggesting that consumption of nuts, a staple in the Mediterranean diet, is associated with reduced risks for cancer development and cancer-related deaths.

Based on the pooled results of eight prospective studies, the meta-analysis by Aune et al. showed that highest vs. lowest nut consumption was associated with a significant 15% reduction of total cancer incidence [8]. The most recent and comprehensive meta-analysis by Naghshi et al. comprised 51 epidemiological studies and reported that the summary effect size for risk of cancer, comparing extreme categories of total nut consumption, was similar to that described by Aune et al. [8], with an RR of 0.86 (95% CI, 0.81, 0.92) [45]. In the dose–response analysis, each 5-g/d increase of total nut consumption was associated with 3%, 6%, and 25% lower risks of overall, pancreatic, and colon cancers, respectively. Of note, this inverse dose–response relationship between nuts and incident cancer was not significant for peanuts and peanut butter consumption [45]. This could be due to different nutrient composition in peanuts vs. tree nuts [1]. Peanut butter, though comprised predominantly of ground peanuts that have been roasted, generally also contains additives such as sugar, salt and hydrogenated oils that may hamper its health benefits [46], as shown for CVD. Moreover, when peanuts or other nuts are improperly stored, they can be contaminated with aflatoxins, which are potent carcinogens produced by certain molds.

Nuts may also mitigate the increased breast cancer risk associated with alcohol use, as noted in a cohort study that showed reduced risk of benign breast disease, a precursor for breast cancer, with nut consumption, especially among individuals with substantial alcohol use [47].

The Naghshi et al. meta-analysis [45] also evaluated the effects of nuts on deaths due to malignancy and found statistically significant 18%, 8%, and 13% risk reductions in the risk of cancer mortality with the higher intake of tree nuts, peanuts, and total nuts, respectively. Like for the case of incident cancer, no significant association between peanut butter consumption and cancer mortality was ascertained [45]. Many other studies have evaluated the relationship between nut consumption and risk of cancer mortality. An earlier meta-analysis by Zhang et al. in 2020 showed that total nut consumption was associated with a reduced odds ratio (OR) of cancer-related mortality (OR = 0.90; 95% CI, 0.88, 0.92) [48]. In this meta-analysis, statistically significant inverse associations between nut consumption and cancer site occurrence were present for colorectum, stomach, pancreas, and lung.

In summary, a consistent body of evidence, albeit based exclusively on observational studies, suggests that consumption of nuts may modestly reduce cancer incidence and cancer-related deaths. These benefits appear to be most significant for tree nuts, less so for peanuts, and nonexistent for peanut butter. More large prospective studies and RCTs are needed to clarify this important issue.

### 5.5. Brain Health

An unwanted consequence of increased lifespan and associated population aging in recent decades is a growing number of elderly individuals at risk of neurodegenerative disorders, particularly Alzheimer’s disease—the most common type of dementia. Given that no effective disease-modifying pharmacologic treatments for mild cognitive impairment, a common harbinger of dementia, or dementia itself are available [49], there is an increasing interest in preventive strategies to implement in preclinical and early stages. Among them, lifestyle modifications, including dietary changes, are being actively investigated and there is incipient evidence that they may forestall cognitive decline and even prevent dementia, particularly in individuals at higher risk [50,51]. Brain oxidative stress and inflammation are currently considered to be causal factors leading to age-related neurodegeneration, a reason why dietary patterns and foods with anti-inflammatory properties are the most promising for improving brain health [52]. Additionally, a close link exists between Alzheimer’s disease and vascular pathology, and there is evidence that treatment of CV risk factors contributes to maintain neuronal integrity and prevent cognitive dysfunction [53].

Nuts are rich in neuroprotective nutrients such as PUFAs and polyphenols and their increased consumption benefits vascular function and is consistently associated with reduced rates of CVD, therefore it can be predicted that they might also beneficially influence cognition and overall brain health. Although data from cohort studies relating nut consumption to dementia outcomes are lacking, evidence is accumulating on the potential of nuts to improve cognitive function. A recent systematic review synthetized data from 14 epidemiological studies and eight RCTs assessing effects of nut-enriched diets on cognitive function [54]. While some epidemiological studies showed a positive association, the quality of the evidence was low because nine studies were cross-sectional or case–control and only five were prospective. Nevertheless, studies targeting populations at higher risk of cognitive decline tended to have favorable outcomes. Notably, studies that specifically addressed the association between walnut consumption and cognitive performance had more homogeneous results, as out of six walnut studies, including two RCTs, only one failed to find a positive association. This may be due to the highly bioactive nutrients of walnuts previously mentioned in reference to T2D risk [41]. Indeed, many studies using walnuts in experimental models of brain aging and neurodegeneration have consistently uncovered beneficial effects [55]. A large population-based prospective study published after that systematic review, the Singapore Chinese Health Study, supports the cognitive benefit of nuts, as nut consumption at midlife was associated with a dose-dependent reduction in risk of cognitive impairment 20 years later [56]. Interestingly, unsaturated fatty acid intake mediated close to 50% of the beneficial effect, pointing to the fatty acid composition of nuts as relevant in improving cognition.

Late-life depression is a common psychiatric disorder that compromises the quality of life of affected individuals. Depression is also a risk factor for cognitive decline, where chronic inflammation contributes to its pathophysiology, as is the case with neurodegenerative disorders [57]. For a similar reason, depression is also a risk factor for CHD, although the association is bidirectional [58]. Consequently, nuts can be postulated to have a salutary effect on depression. The epidemiologic evidence, however, is scanty and of suboptimal quality. Nut consumption was reported to benefit depressive symptoms in a large cross-sectional study of Chinese adults [59]. In another cross-sectional report from the US, nut consumers, and particularly walnut consumers, disclosed lower depression scores than subjects who were not consuming nuts, and this beneficial effect was more pronounced in women [60]. In that study, food consumption was assessed only via 24 h diet recalls, which can provide strong evidence for frequently consumed foods but, unless repeated, are much weaker for sporadically consumed foods such as nuts. In the Invecchiare in Chianti study, an Italian prospective investigation of 1058 adults followed for up to 9 years with repeated measurements of diet and depression scores, no association between consumption of nuts and depressive symptoms was observed [61].

Brain-beneficial nutrients contained in nuts, such as PUFAs and polyphenols, support their potential to delay cognitive decline, as evidenced in a few prospective studies and RCTs. Furthermore, other nut components such as phytomelatonin, phytosterols, antioxidant tocopherols, and folic acid may also support neurological health and cognitive wellness. Clearly, more well-designed prospective studies and RCTs, preferably conducted in individuals at high risk or with early dementia stages, are warranted to uncover the full potential of nuts to counteract cognitive decline.

### 5.6. All-Cause Mortality

While the major focus of epidemiological research with nuts has been CVD, many large population-based prospective cohort studies conducted globally have examined associations of exposure to nuts or nut components such as peanut butter with all-cause mortality. The latest meta-analysis by Chen et al. [62] synthetized data from 18 prospective studies and obtained a summary RR for high compared with low nut consumption of 0.81 (95% CI, 0.78, 0.84) for all-cause mortality. When data for total nuts and tree nuts and peanuts were analyzed separately, the RR estimates were similar for the three nut categories. Only two studies examined peanut butter separately from peanuts and their combined RR for all-cause mortality was 0.89 (95% CI, 0.80, 0.99). In dose–response analysis, the RR for all-cause mortality per one additional serving of total nuts per week was 0.96 (95% CI, 0.94, 0.97). Interestingly, dose–response analyses revealed nonlinear inverse associations between nut consumption and mortality, with risk reduction leveling off at consumption of approximately 3 servings/week (equivalent to 12 g/d), which suggests that maximum benefit on survival may be achieved with relatively low doses of nuts.

That low nut doses relate to lower overall mortality is underlined in a recent report from a large (*n* = 31,552) population-based prospective Japanese study, whereby higher compared with lower nut consumption was associated with reduced all-cause mortality (HR = 0.85, 95% CI, 0.75, 0.96) in men (not in women), in spite of an average consumption of only 1.8 g/d, peanuts accounting for 80% of total nuts [31]. Additionally, in spite of a similarly low average nut consumption in Korea, a recent large cross-sectional population survey relating dietary factors to all-cause and cause-specific mortality using a comparative risk assessment analysis found that a sizable proportion of deaths was related to low consumption of nuts [63]. The results of a recent very large (*n* = 566,398) population-based prospective study in the US with a median follow-up of 15.5 years support the inverse association between higher nut consumption and total mortality (HR = 0.78; 95% CI, 0.76, 0.81); in contrast consumption of peanut butter was not associated with lower risk of mortality [27]. Finally, the recent report from the PURE study described a significant reduction in total mortality (HR = 0.77; 95% CI, 0.69, 0.87) for highest (≥120 g/week) versus lowest (<30 g/month) nut consumption [28]. In PURE, tree nut consumption was associated with a decreased risk of mortality, whereas peanut consumption disclosed a nonsignificant trend towards a lower mortality risk.

Nut consumption in relation to mortality has also been examined in a large prospective cohort of individuals with T2D. In the NHS and HPFS report on nuts and mortality among 16,217 men and women with diabetes, higher vs. lower nut consumption was associated with a significant 31% reduction in all-cause mortality [26]. When assessed separately, consumption of tree nuts and peanuts related to 33% and 20% lower mortality risks, respectively. Overall, the findings from prospective studies consistently point to an inverse association of nut consumption with all-cause mortality, with an average of 1 in 5 deaths prevented or delayed by nut consumption at moderate levels.

### 5.7. Intermediate Markers: Adiposity, Lipids, Blood Pressure, Glycemic Control, Endothelial Function, and Inflammation

#### 5.7.1. Adiposity

The steady increase in the prevalence of overweight/obesity worldwide is a major public health problem. Due to the high energy density of nuts, increased body weight with long-term consumption has been an underlying concern. Yet, to the contrary, a growing body of epidemiological evidence suggests that daily nut consumption is a potentially effective strategy in the primary prevention of obesity [64].

A 2014 review of epidemiological and RCT data concluded that evidence was lacking on the common assertion that regular consumption of nuts increased adiposity [10]. This was confirmed in a recent network meta-analysis of 105 RCTs comparing the effects of diets enriched in various tree nuts and peanuts vs. control diets on body weight, BMI, waist circumference (WC), and percent body fat [64]. No significant increase was observed in any adiposity measures with any of the nuts, except for hazelnut-rich diets, which raised WC. On the other hand, results of pairwise comparisons between different nuts indicated that almond diets reduced WC compared to control diets; walnuts also reduced WC compared to pistachio, hazelnut and mixed nuts-enriched diets. In subgroup analyses considering only RCTs specifically designed to assess the weight loss effects of nut consumption, almonds were associated with reduced BMI and walnuts with reduced percent body fat. Importantly, among overweight and obese study subjects, those who consumed nut-enriched diets experienced greater weight loss, reduced BMI and lower WC compared with their counterparts who consumed a nut-free isocaloric control diet (Figure 3).

Another recent meta-analysis of nut-feeding trials examined whether providing or not dietary substitution instructions to participants (recommending foods to be replaced by the nuts or just advising to eat the nuts on top of the usual diet) influenced adiposity changes [65]. The results showed the same absence of weight, BMI or WC changes for the two categories of studies. 

A very large prospective study involving the three Harvard cohorts of the NHS-I, NHS-II, and HPFS assessed the association between changes in consumption of total and specific nuts per 4 y intervals and weight changes over 20–24 y of follow-up [66]. Increases in nut consumption, per 0.5 servings/d (14 g), were significantly associated with less weight gain per 4 y interval: −0.19 kg (95% CI, −0.21, −0.17) for total nuts, −0.37 kg (95% CI. −0.45, −0.30) for walnuts, −0.36 kg (95% CI, −0.40, −0.31) for other tree nuts, and −0.15 kg (95% CI, −0.19, −0.11) for peanuts. An increase in consumption of total nuts, per 0.5 servings/d, was associated with a modest but significant 3% lower risk of becoming obese, while a similar increase in consumption of walnuts and other tree nuts was associated with a 15% and 11% lower risk of developing obesity, respectively. Increasing peanut consumption, however, was not associated with reduced obesity risk.

Thus, both epidemiological and RCT data point to a slightly beneficial effect of nut consumption on adiposity rather than a harmful effect. That regularly eating a highly energy-dense food does not promote a positive energy balance is of particular interest. Several mechanisms underly the associations between nut consumption and lower risk of weight gain [10]. Nuts require considerable effort at mastication and chewing, and their high fat and fiber content can delay gastric emptying, increase satiety, suppress hunger and promote fullness. The fiber in nuts also increases binding of fatty acids in the gut, leading to greater fecal fat excretion. Similarly, the efficiency of energy absorption from nuts is reduced due to incomplete mastication and encasement of fat within unbroken cell walls in nut particles, hampering the bioaccessibility of fat from nuts in the gastrointestinal tract, with ensuing increases in fecal energy (fat) loss. Finally, there is evidence that the high unsaturated fat levels in nuts enhance fatty acid oxidation and increase thermogenesis and resting energy expenditure, which may also mitigate weight gain.

#### 5.7.2. Blood Lipids

Since the landmark RCT of Sabaté et al. demonstrating the cholesterol-lowering effect of a walnut diet [3], the effects of diets enriched with different nuts on blood lipids and lipoproteins have been examined in many RCTs [35].

To date, the 2015 meta-analysis of Del Gobbo et al. [9] is the most comprehensive. It reviewed 61 intervention trials (42 randomized and 19 non-randomized) lasting from 3 to 26 weeks designed to assess the effects of tree nuts on the blood lipid profile. All trials provided the study nuts to participants rather than simply giving advice to procure the nuts by themselves. Nut consumption (per serving/d) significantly decreased total cholesterol (−4.7 mg/dL), LDL-cholesterol (−4.8 mg/dL), and triglycerides (−2.2 mg/dL), but had no effect on HDL-cholesterol. Walnuts, followed by almonds and pistachios, were the nuts most frequently studied. The LDL-cholesterol lowering effect was dose related in a non-linear fashion, with stronger effects at doses of 60 g/d (approximately 2 servings), while triglyceride lowering had a linear dose–response. There was no heterogeneity by nut type or quality of the control diet. These authors reanalyzed the data as a function of the phytosterol content of nuts in each study and demonstrated that the phytosterol dose was strongly related to the observed LDL-cholesterol reduction, although this association was driven by the total nut dose [21].

An earlier analysis with pooled individual data from 21 RCTs indicated that, for an average consumption of 67 g/d of tree nuts or peanuts (two servings, approximately 20% of energy), the mean estimated reduction of LDL-cholesterol was 10 mg/dL (7%) [67]. Nuts had no significant effect on serum triglycerides, except in participants with triglycerides >150 mg/dL, in whom a significant 10.2 mg/dL reduction was observed. Importantly, there was a clear dose–response in LDL-cholesterol lowering. The statistical power of this pooled analysis allowed detection of differential responses by baseline LDL-cholesterol level (greater response with higher values) and BMI (greater response with lower BMI) (Figure 4). The mean 10% LDL-cholesterol reduction with 2 servings/d of nuts in hypercholesterolemic individuals is similar to that described for functional foods fortified with plant sterols/stanols [68], which epitomizes the nutraceutical properties of nuts as cholesterol-lowering foods. Recently, in a network meta-analysis of 66 RCTs comparing the effects of 10 common food groups (refined grains, whole grains, nuts, legumes, fruits and vegetables, eggs, dairy, fish, red meat, and sugar-sweetened beverages) on cardiometabolic outcomes, nuts were ranked as the best food group at reducing LDL-cholesterol [69].

The lipid effects for individual nut types have been examined in meta-analyses of RCTs using walnuts (24 studies) [70], almonds (27 studies) [71], pistachios (11 studies) [72], hazelnuts (3 studies) [73], and cashews (3 studies) [74]. All individual nuts except cashews reduced LDL-cholesterol to a similar extent than reported for total nuts in the mentioned systematic reviews [9,67], but cashews had no effect, which may be due to the low number of RCTs analyzed. Finally, a recent network meta-analysis of 34 RCTs of these five nuts for lipid outcomes used analyses based on the surface under the cumulative ranking curves and concluded that diets enriched in pistachios and walnuts were best for lowering LDL-cholesterol and triglycerides compared with the other nut-enriched diets included in the study [75].

#### 5.7.3. Blood Pressure

The effects of nuts on office BP have been reported in many RCTs [9,35]. BP changes were a secondary outcome in the 2015 meta-analysis of Del Gobbo et al. [9], and no effect of nut-enriched diets on either systolic BP (SBP) or diastolic BP (DBP) was found. A 2015 meta-analysis including 21 RCTs of nut diets by Mohammadifard et al. [76] focused on BP changes. Results showed that diets supplemented with nuts had no effect on BP overall, except in individuals without T2D, who disclosed a weighted mean difference (WMD) in SBP of −1.29 mm Hg; (95% CI, −2.35, −0.22). In sub-analyses stratified by nut types, only diets enriched in pistachios resulted in a significant BP reduction, with a WMD of −1.82 mm Hg (95% CI, −2.97, −0.67) for SBP and of −0.80 mm Hg (95% CI, −1.43, −0.17) for DBP, while mixed nuts reduced only DBP, with a WMD of −1.19 mm Hg (95% CI, −2.35, −0.03).

Data on BP changes for specific nut types have also been reported. Thus, the 2018 meta-analysis by Guasch-Ferré et al. [70] of 24 RCTs focused on CV risk factor changes with walnut-enriched diets reported no effect on BP. A recent meta-analysis of 16 RCTs examining the effects of almonds on BP showed no differences for SBP between almond and control diets, but pooled analyses revealed a significant reduction of DBP by almond diets (WMD = −1.30 mm Hg; 95 % CI, −2.31, −0.30) [77]. A meta-analysis of 13 RCTs using pistachios for outcomes of CV risk factors by Asbaghi et al. [78] indicated a significant reduction of SBP (WMD = −2.12 mm Hg; 95 % CI, −3.65, −0.59), which supports the findings of the Mohammadifard et al. meta-analysis [76], although no effect on DBP was found. The meta-analysis of 3 cashew RCTs by Jalali et al. [74] also reported a significant reduction of SBP (WMD = −3.39 mm Hg; 95% CI, −6.13, −0.65), without changes of DBP. 

The evidence on the effects of nuts on BP outcomes is inconsistent and, in general, does not support a relevant lowering effect, which contrasts with the epidemiological findings of an association of nut diets with a lower risk of incident hypertension, consistent across different meta-analyses [35,36]. Reasons for the failure of RCTs to detect BP changes with nut-enriched diets may be low statistical power (most RCTs included less than 50 participants), short duration of the intervention, exclusive use of office BP measurements, and the fact that they were usually a secondary outcome of lipid-focused trials, hence were not powered to detect changes in BP. Recently, the 2-year effects of a walnut diet on both office BP and 24-h ambulatory BP (the gold standard of BP measurements) in the Walnuts and Healthy Aging (WAHA) RCT conducted in 236 older individuals were reported [79]. The results showed that, compared with a control diet, a diet supplemented with walnuts at ≈15% of energy resulted in lower office SBP (−4.61 mm Hg) in the whole cohort and reduced 24-h ambulatory SBP (−8.5 mm Hg) in hypertensive participants. No changes in diastolic BP were observed. During the trial, participants in the walnut group required less uptitration of antihypertensive medication and had better overall BP regulation than controls. The WAHA trial overcomes the limitations of prior RCTs concerning BP effects of nut diets and shows a beneficial effect of long-term walnut consumption on SBP.

#### 5.7.4. Glycemic Control

Acute feeding studies have shown that nuts consumed with carbohydrate-rich foods having a high glycemic index reduce postprandial glucose responses in comparison with consumption of the same foods alone in both normoglycemic individuals and those with T2D [80,81], which suggests that nuts may be useful in glycemic control. The evidence from RCTs, however, is mixed. A recent meta-analysis of 40 RCTs with a median duration of 3 months concluded that consumption of tree nuts or peanuts had modest favorable effects on the homeostasis model assessment of insulin resistance (HOMA-IR) (WMD = −0.23) and fasting insulin (WMD = −0.40 μIU/mL), but not on fasting blood glucose or hemoglobin A1c [82]. Subgroup analyses showed similar results whether the study subjects were healthy individuals or those with prediabetes or T2D.

A meta-analysis of 16 RCTs that assessed effects of walnut diets on biomarkers of glycemic control failed to find any benefit [83]. Likewise, a recent in-depth narrative review of almonds and health outcomes based on findings of 64 RCTs and 14 meta-analyses and/or systematic reviews concludes that almonds have inconsistent and/or insignificant beneficial effects on glycemic control [81].

#### 5.7.5. Endothelial Function

The endothelium plays a central role in arterial health and throughout all stages of atherosclerosis. Endothelial function can be viewed as an integrative biomarker of the overall harmful effects of CV risk factors on the arterial wall, a reason why endothelial dysfunction is an independent predictor of future CVD events [84]. Endothelial dysfunction is characterized by a decreased bioavailability of NO and increased expression of pro-inflammatory cytokines and cellular adhesion molecules and can be evaluated non-invasively by several methods; flow-mediated dilation (FMD) measured by brachial artery ultrasound is considered the most sensitive and accurate in assessing endothelial function [85].

Two meta-analyses have summarized results of RCTs testing nut diets for effects on FMD [86,87]. The meta-analysis of Neale et al. [86] examined RCTs of nut diets providing data on inflammatory molecules, but also regarding effects on endothelial function. FMD was explored in nine strata (five testing the effects of walnuts) from eight RCTs, resulting in significant improvements in FMD of the nut versus the control diets (WMD = 0.79%; 95% CI, 0.35, 1.23). When subgroup comparisons were made according to nut type, only the walnut interventions resulted in improved FMD. The meta-analysis of Xiao et al. [87] of 10 RCTs was focused exclusively on effects of nuts on FMD. The pooled estimates showed that nut consumption significantly improved FMD (WMD = 0.41%; 95% CI, 0.18, 0.63). Again, subgroup analyses indicated that only walnut interventions improved FMD. Walnuts are particularly rich in ALA, polyphenols, arginine (the precursor of NO, the endogenous vasodilator) and other bioactives, as reviewed [41], which may explain their differential effects on endothelial function. However, almonds may also improve endothelial function, as shown by a recent 6-wk RCT that tested almond snacks (about 2 servings/day) versus control snacks (muffins) in adults at above-average CV risk for effects on FMD, among other cardiometabolic risk variables [88]. The results showed a noticeable increase in FMD by almonds (WMD = 4.1%; 95% CI, 2.2, 5.9), much higher than that reported in the cited meta-analyses [87,88]. No effects on BP were observed despite the use of 24 h ambulatory BP monitoring.

A recent review analyzed 16 nut intervention trials using noninvasive techniques other than FMD to assess vascular function, such as pulse wave velocity, pulse wave analysis, digital volume pulse, impedance cardiography, and peripheral arterial tonometry [89]. The results were mixed, with only 6 out of 16 studies showing improved vascular function ensuing nut diets.

#### 5.7.6. Inflammation

Chronic non-communicable diseases, such as atherosclerosis with major CV events, obesity, T2D, neurodegenerative disorders, cancer, and auto-immune diseases are characterized by a state of low-grade inflammation, which plays a central role in disease progression and perpetuation. Changes in this inflammatory state can be identified by determination of circulating biomarkers of inflammation, including C-reactive protein (CRP), tumor-necrosis factor-alpha (TNF-α), interleukin-6 (IL-6), E-selectin, and adhesion molecules intercellular adhesion molecule (ICAM)-1 and vascular cell adhesion molecule (VCAM)-1, all of which have garnered much interest in CV risk prediction [90]. The critical role of chronic inflammation in CVD has been substantiated recently by landmark RCTs demonstrating that interventions selectively targeting inflammation can improve clinical outcomes in patients with atherosclerosis [91].

The effects of nut-enriched diets on soluble inflammatory biomarkers have been investigated in many RCTs, usually as secondary outcomes, therefore not powered to detect changes in these outcomes [35]. A 2017 meta-analysis of 32 RCTs by Neale et al. [86] concluded that nut interventions induced no significant changes in inflammatory markers, including CRP, TNF-α, IL-6, ICAM-1 and VCAM-1, or in the anti-inflammatory biomarker adiponectin. A 2018 meta-analysis of 23 RCTs by Xiao et al. [92] showed that nut consumption reduced ICAM-1 (WMD = −0.17; 95% CI, −0.32, −0.03), but had no consistent effects on CRP or other soluble inflammatory molecules. Other meta-analyses focused on the lipid effects of total nuts [9] and walnuts [70] and a recent review on the cardiometabolic effects of almonds [81] concur in reporting no significant changes in CRP levels.

While based on results of generally small and short-term RCTs, it appears that consumption of nuts has a negligible impact on inflammatory markers, a recent report from the large, long-term WAHA trial provides a different view. The walnut intervention at ≈15% of energy for 2 years in 634 older participants recruited in two sites, Barcelona, Spain and Loma Linda, California, resulted in significant mean reductions ranging from 3.5% to 11.5% in several inflammatory biomarkers, including granulocyte-monocyte colony stimulating factor, interferon-γ, IL-1-β, IL-6, TNF-α, and E-selectin, but had no effect on CRP, ICAM-1 or VCAM-1 [93]. Thus, high statistical power and a long duration of the interventions might be necessary to uncover the anti-inflammatory effects of nuts. Regardless, these data provide novel mechanistic insight for the benefit of nut (walnut) consumption on CVD risk beyond that of lipid lowering.

### 5.8. Other Health Outcomes

Few prospective studies or single RCTs have examined the effects of nuts on alternative health outcomes, such as gallbladder disease, metabolic syndrome (MetS), non-alcoholic fatty liver disease (NAFLD), physical function, healthy aging, bone health, and reproductive health. There is also incipient evidence that nut diets elicit changes in microbiota.

Two reports from the large prospective cohorts of the NHS in women and the HPFS in men examined the association between frequency of nut consumption and risk of gallstone disease. In the NHS, women consuming ≥5 servings of nuts per week had a significantly lower risk of cholecystectomy (RR = 0.75, 95% CI, 0.66, 0.85) than did those who rarely or never consumed nuts [94], while in the HPFS, men consuming ≥5 servings of nuts per week had a significantly lower risk of symptomatic gallstone disease (RR = 0.70, 95% CI, 0.60, 0.86) compared to those who rarely or never consumed nuts [95]. The results of the two studies suggest that regular nut consumption protects men and women equally against gallstone disease. This beneficial effect is attributable to the richness of nuts in bioactive components capable of influencing intestinal bile acid and cholesterol biology, particularly unsaturated fatty acids, fiber, and non-sodium minerals.

An individual meets diagnostic criteria for MetS when harboring at least three of the following risk factors: increased WC, high triglycerides, low HDL-C, elevated BP, and high fasting blood glucose, and this cluster of risk factors increases the risk of CVD and all-cause mortality beyond the risk imparted by each separate factor [96]. The pathophysiological basis of MetS is insulin resistance, generally linked to central fatness, and as such MetS is an epidemic condition worldwide. Lifestyle changes directed to weight loss and cardiometabolic risk factor control are critical for preventing and treating MetS, thus nut consumption might play a role [97]. A 2014 meta-analysis included 49 RCTs of ≥3 weeks duration reporting effects of nut consumption on at least one criterion of the MetS [98]. Pooled analyses showed a beneficial effect of nuts on MetS via modest decreases in triglycerides and fasting blood glucose. However, it should be noted that a recent meta-analysis found no evidence of benefit of nut diets on blood glucose levels [82], which underlines the limitations of present data on a putative beneficial effect of nuts on MetS. Likewise, the large WAHA trial found no effect of a 2-year walnut-enriched diet on MetS [99].

NAFLD, the accumulation of fat (triglycerides) in the liver in the absence of excessive alcohol intake is the hepatic manifestation of MetS, a prevalent condition globally and a public health concern. NAFLD not only increases risk of CVD, but also of liver cirrhosis and hepatocarcinoma [100]. Like in MetS, abdominal obesity and T2D are major drivers of NAFLD, and its primary treatment consists of lifestyle and dietary changes directed at weight loss. The favorable effects of nuts on body weight, glycemic control and CVD risk would predict a beneficial effect in NAFLD, and a few prospective studies have suggested that increased nut consumption is associated with a lower incidence of NAFLD, as recently reviewed [101]. Nuts may contain the carcinogenic agent aflatoxin, a fungal metabolite and mycotoxin that can contaminate improperly stored nuts and other seeds. This has been a reason of concern for patients with NAFLD due to their increased risk of liver cancer. However, in Western countries, where aflatoxin contamination of crops is rare due to strict regulations, health benefits provided by increased nut consumption likely outweigh the risks associated with chronic increases in aflatoxin exposure. This may not be the case in countries known for high rates of aflatoxin contamination of peanuts, like Indonesia [101].

There is evidence from a single prospective study conducted in Spain, where nut consumption is rather high, that it may lower the risk of impaired agility/mobility and increase overall physical function in older individuals [102], while another report from the NHS cohort suggests that consumption of total nuts and, particularly, walnuts is associated with healthy aging, i.e., survival beyond 65 years with no chronic diseases, no memory impairment, no physical disabilities, and intact mental health [103].

Dietary components are important for providing crucial constituents for bone health and regulating cellular metabolism within bone [104]. Nuts might promote bone health because they are rich sources of antioxidant, anti-inflammatory flavonoids, and calcium. Resveratrol is a stilbene-type polyphenol present in some nuts that is a powerful activator of the longevity-linked sirtuin-1 molecule, which regulates processes related to longevity, including apoptosis, DNA repair and energy expenditure [105]. Nevertheless, there are no data from prospective studies relating nut consumption to bone health, while RCTs are limited. A single small RCT assessed the effects of an ALA diet sourced from walnuts and flaxseed oil in comparison with an average American diet and a linoleic acid-rich diet on bone turnover, assessed by serum concentrations of N-telopeptides and bone-specific alkaline phosphatase [106]. N-telopeptide levels were significantly lower following the ALA diet relative to the average American diet, suggesting that plant sources of dietary n-3 PUFA may have a protective effect on bone metabolism via decreased bone resorption. In summary, the evidence on the efficacy of nuts to promote bone health is very limited. Both well-powered prospective studies and RCTs are warranted to examine this important issue.

Concerning reproductive health, in the last decades there has been a steady increase in infertility worldwide, in great part related to declining semen quality. Exposure to environmental toxins, smoking, and unhealthy diets are believed to underlie impaired spermatogenesis [107,108]. Two RCTs have tested nut diets for outcomes of semen quality. Robbins et al. [109] randomized 117 healthy men to consume 75 g of walnuts/d for 12 weeks on top of their usual Western-style diet or usual diet alone and found improved sperm vitality and motility after the walnut diet, but no changes in total sperm count. The FERTINUTS trial [110] was a 14-wk RCT involving 119 healthy men 18–35 y-old that assessed the effects on various sperm parameters of 60 g/day of mixed nuts (30 g walnuts, 15 g almonds, and 15 g hazelnuts) in the context of a Western-style diet vs. the same diet without nuts. Compared to the control group, the nut group showed significant improvements in total sperm count and vitality, motility and morphology. Nuts appear to improve male fertility, but clearly more research is needed.

Another prevailing pathology related to men’s sexual heath is erectile dysfunction, a condition in which endothelial dysfunction at the level of penile vasculature is causal in the failure to initiate and/or maintain an erection [111]. Erectile dysfunction is intimately linked to CV risk factors and associated with an increased incidence of CVD [112]. Given that nut consumption is associated with reduced CV risk and improved endothelial function, in part due to their content in arginine, the precursor of the endogenous vasodilator NO, it is plausible that nut diets would benefit erectile function. Indeed, in folk medicine nuts (particularly cashews and walnuts) are promoted as aphrodisiacs and a remedy for impotency. A single RCT, a secondary analysis of the FERTINUTS trial [110], tested the effect of mixed nuts on erectile function in healthy young men [113]. Compared to the control group, small but significant increases in self-reported orgasmic function and sexual desire, but not erectile function, were observed following the nut intervention. There were no between-group differences in changes of peripheral concentrations of NO and E-selectin. Well-powered RCTs conducted in individuals with an objective diagnosis of erectile dysfunction are necessary to reach definitive conclusions on the efficacy of nuts to help men with this prevalent and troublesome pathology.

An expanding area of clinical research is the intestinal microbiome, which is primarily controlled by the nutritional quality of the diet and is believed to play a major role in a vast array of biological functions [114]. Colonic microbiota can be modulated by different lifestyle and dietary factors and impact the risk of developing obesity, T2D and other cardiometabolic diseases, as well as infectious diseases. Nuts have been suggested to have a prebiotic effect (that conferred by a substrate selectively used by the host microorganisms translating into a health benefit) on the gut microbiome [115]. The non-bioaccessible components of nuts (fiber, polymerized polyphenols and fat contained within undigested cell walls in incompletely masticated nut particles) make up a rich supply of nutrients to the intestines for feeding the microbes residing there. The field of nuts and microbiota is still at an early stage, but a recent comprehensive meta-analysis synthesized data from nine RCTs investigating almonds (*n* = 5), walnuts (*n* = 3) and pistachios (*n* = 1) for effects on fecal bacterial diversity [116]. Nut consumption increased the relative abundances of the genera *Clostridium*, *Lachnospira* and *Roseburia*, which are considered beneficial because they produce butyrate, a short-chain fatty acid critical in nourishing the intestinal epithelium and maintaining its integrity. Nut consumption had little overall impact on bacterial diversity, a metric considered as positive for health, except for a marginal enhancement from almond consumption, which could be explained by the particular matrix of almonds and small cell walls limiting fat availability for digestion, but increasing fat delivery to the colon and thus feeding the microbiota [81]. The overall meta-analytical evidence of a modulatory effect of microbiota by nuts is weak because microbial determinations were a secondary outcome in most RCTs, which were not powered to detect changes of this outcome. Nut effects on microbiota is a relevant topic for future research.

### 5.9. Health Effects of Nuts in the Predimed Trial

The landmark PREDIMED trial targeted both the effects of nut consumption on intermediate cardiometabolic markers and clinical outcomes, such as CVD and T2D, among others. The PREDIMED study [23] was a multicentric, parallel group, nutrition intervention RCT for the primary prevention of CVD. It was conducted in Spain and enrolled 7447 men and women aged 55–80 years at high risk of CVD but no CVD at recruitment. Participants were allocated to three study arms: two Mediterranean diets, supplemented with either extra-virgin olive oil (50 mL or more/day) or mixed nuts (30 g/day: 15 g walnuts, 7.5 g almonds, and 7.5 g hazelnuts), or control diet (advice on a low-fat diet) and followed for 5 y. The supplemental foods (olive oil and raw, unpeeled nuts) were delivered periodically to participants in the corresponding groups. Registered dietitians delivered the interventions at quarterly individual visits and group sessions separate for each group. As PREDIMED intended to assess the effects of the nutrition intervention alone, the diets were energy-unrestricted and increased physical activity was not promoted. The primary end point was a composite of major CVD events (non-fatal myocardial infarction, non-fatal stroke, and CV death). An event adjudication committee, whose members were blinded to group allocation, was responsible for event ascertainment. Attesting to the high CV risk of participants, the mean age was 67 years and the mean BMI was 30 kg/m^2^, almost one-half had T2D, two-thirds had dyslipidemia, and 4 out of 5 had hypertension. Since its inception in June 2003, the trial has generated a steady stream of data on the beneficial health effects of Mediterranean diets enriched with either of the supplemental foods, culminating with the publication of results on the primary CVD outcome, demonstrating a ≈30% reduction with the two Mediterranean diets compared with the control diet [23]. The incidence of myocardial infarction was reduced non significantly with the two Mediterranean diets. The main results concerning the Mediterranean diet enriched with nuts are summarized below.

The most striking result was that incident stroke, a component of the PREDIMED main outcome, was significantly reduced by 45% (HR = 0,55; 95% CI, 0.35, 0.86] in the group allocated the Mediterranean with nuts compared to the control group (Figure 5) [23]. Regarding other hard cardiometabolic outcomes, the Mediterranean diet with nuts resulted in a 49% reduction (HR = 0.51; 95% CI, 0.32, 0.83) in the incidence of peripheral artery disease [117] and a non-significant 18% reduction (HR = 0.82; 95% CI, 0.61, 1.10) in incident T2D [118]. In participants with MetS at baseline (*n* = 3392), the nut-supplemented diet resulted in a 28% (HR = 1.28; CI 1.08, 1.51) higher probability of reversion of MetS compared with the control diet, and this beneficial effect was driven mainly by reduced WC [119]. The risk of heart failure, a secondary outcome of the trial, was unaffected by the Mediterranean diets [120]. A post hoc analysis revealed no effect of the nut-supplemented diet on incident atrial fibrillation (HR = 0.89; 95% CI, 0.65, 1.20) [119]. The trial was not powered to examine mortality risk; however, when considered as an observational cohort, nut consumption was associated with a significantly reduced risk of all-cause mortality: compared to non-consumers, participants consuming nuts >3 servings/week (32% of the cohort) had a 39% lower mortality risk (HR = 0.61; 95% CI, 0.45, 0.83) [121]. A similar protective effect against CVD and cancer mortality was observed.

Concerning intermediate outcomes, data from the full PREDIMED cohort showed a stable body weight, but WC (which tends to increase with age in older populations) increased less in the Mediterranean diet with nuts group, with and adjusted difference in 5-y changes of –0.92 cm (95% CI, –1.60, –0.24) compared with the control group [122]. These results provide first-level evidence that an ad libitum Mediterranean diet high in fat because of supplementation with nuts does not promote weight gain or visceral adiposity. Data from PREDIMED sub-studies revealed beneficial changes of the Mediterranean diet with nuts on: blood lipids and fasting glucose [122,123]; office BP, insulin resistance and soluble inflammatory markers (except CRP) [122]; BP as assessed by 24 h ambulatory monitoring, with 1 y changes of nearly −4 mm Hg for SBP and −2 mm Hg for DBP compared with the control diet, remarkable given that most participants were hypertensive and received standard anti-hypertensive medications [123]; and carotid plaque regression compared with progression in the control group after intervention for 2.4 y [124]. In another sub-study with 334 participants, a comprehensive neuropsychological test battery was administered at baseline and after a mean follow-up of 4.1 y. The results showed that, compared with the control diet, both Mediterranean diets resulted in delayed age-related cognitive decline, while the nut-supplemented diet performed better in the memory domain [125]. This PREDIMED sub-study is the first RCT demonstrating that a dietary pattern enhances cognitive function.

It must be emphasized that the PREDIMED interventions were meant to improve the overall diet, but the major between-group differences in food consumption were for the provisioned supplemental foods. It follows that nut consumption was probably responsible for most of the observed benefits in the Mediterranean diet with nuts group. The PREDIMED results illustrate the remarkable potential of nuts and other healthy foods such as extra-virgin olive oil to beneficially impact health outcomes. Given the age of PREDIMED participants, an important lesson of the trial is that it is never too late to change dietary habits to improve CV health and brain function.

## 6. Conclusions

Nuts, by virtue of their beneficial nutrients and phytochemicals, appear to bestow favorable and wide-ranging health dividends. The PREDIMED RCT showed a protective effect of nuts against CVD. Other RCTs have demonstrated that nuts lower LDL-cholesterol concentration, reduce insulin resistance and improve vascular reactivity. Epidemiological studies report largely congruent findings indicating that nut consumption is associated with lower risks for CVD, total mortality, atrial fibrillation, hypertension, and cancer. Habitual nut consumption does not promote obesity and may even result in less weight gain over time, particularly among individuals with overweight/obesity. Table 3 summarizes the main beneficial effects of nuts on health outcomes.

The cumulative scientific evidence indicates that nuts are one of the most wholesome and nutritious foods in the usual diet, but which nuts are best for health? The three nuts that were supplied and consumed by the participants in the Mediterranean diet plus nuts arm of the PREDIMED study were walnuts, almonds and hazelnuts. Thus, these are the only three nuts with first level evidence for improving CV outcomes in the context of a RCT.

However, the nutrient profiles of other nuts make them excellent dietary options as well. For example, Brazil nuts are especially rich in selenium, pecans and peanuts are great sources of polyphenol antioxidants, pistachios are particularly high in carotenoids, tocopherols and phytosterols, and macadamias are replete with monounsaturated fats and flavonoids. Consequently, consuming a mixture of nuts, aiming for a daily dose of at least 30 g/d, is ideal for optimizing health.

As nuts are naturally high in non-sodium minerals and virtually devoid of sodium, lightly salted nuts are still a healthy low-salt snack that many people find more palatable than unsalted nuts. The inner peel between the shell and the nut is rich in polyphenols. Given that the peel and its polyphenols are lost when nuts are roasted, raw, unpeeled nuts are generally the healthiest and those that can be rightly considered as the natural food with most pluripotential nutraceutical properties. It is noteworthy that with the choice of a single wholefood we can positively impact multiple cardiometabolic risk factors, promote healthy aging, and live longer [126]. Regular nut consumption is an indispensable component of any healthy, plant-based dietary pattern.

## Figures and Tables

**Figure 1 nutrients-13-03269-f001:**
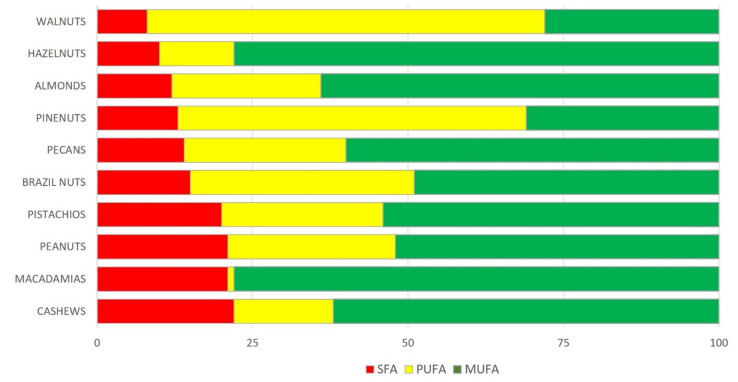
Percent fatty acid profile of common nuts.

**Figure 2 nutrients-13-03269-f002:**
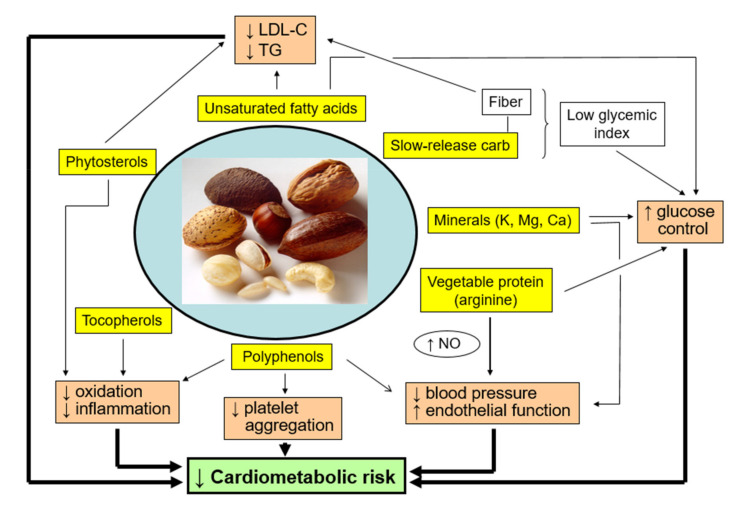
Schematic representation of the effects of nuts on risk of cardiometabolic diseases mediated by their main bioactive nutrients and phytochemicals (yellow boxes), which synergize to positively influence metabolic and vascular physiology pathways (thin arrows and orange boxes). The net effects on intermediate markers of CV risk are lowering of blood cholesterol, improved glycemic control, decreased blood pressure, improved vascular reactivity, and anti-inflammatory actions. Crucially, clinical trials of nuts have demonstrated all such effects. The overall result is reduced cardiometabolic risk (thick arrow connections), as observed in many prospective cohort studies and proven in the PREDIMED trial. Abbreviations: Ca, calcium; K, potassium; LDL-C, LDL-cholesterol; Mg, magnesium; NO, nitric oxide; TG, triglycerides. ↑: increase, ↓: decrease.

**Figure 3 nutrients-13-03269-f003:**
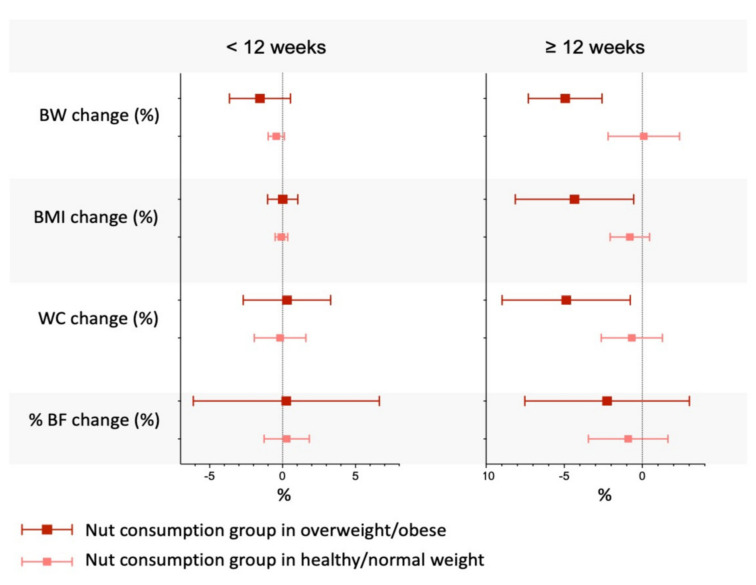
Percentage change for adiposity outcomes in the healthy/normal weight groups vs. the overweight/obesity group with regard to the length of time following the nut interventions in 105 RCTs. Reproduced from reference [64], with permission. BW: body weight; BMI: body mass index; WC: waist circumference; % BF: body fat percentage.

**Figure 4 nutrients-13-03269-f004:**
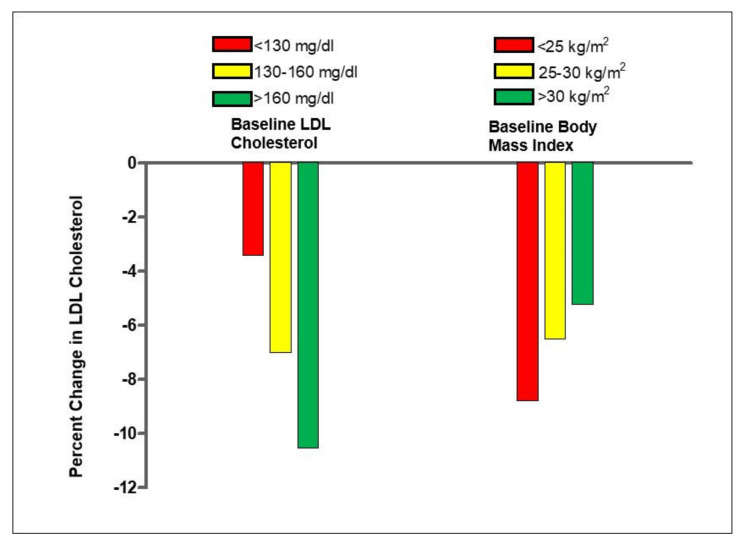
LDL-cholesterol responses to nut diets by baseline LDL-cholesterol and BMI. Data obtained in a pooled study of 25 nut RCTs (adapted from ref. [67] with permission).

**Figure 5 nutrients-13-03269-f005:**
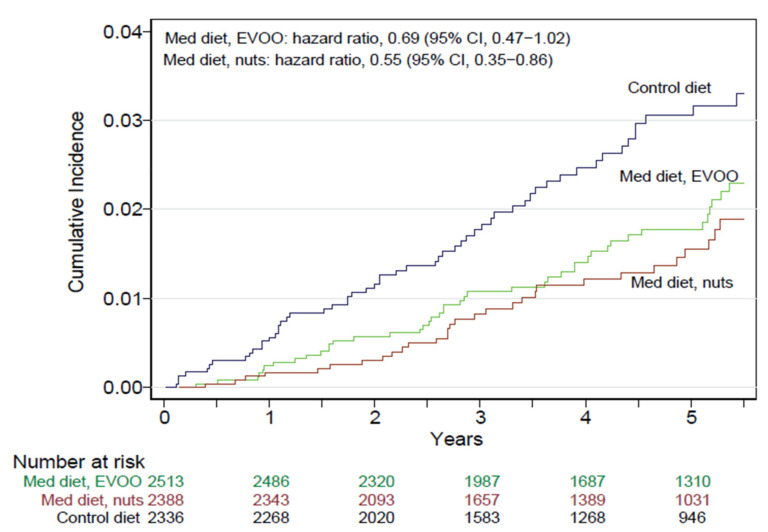
Cumulative incidence of stroke by intervention group in the PREDIMED trial [23]. Copyright © (2018) Massachusetts Medical Society. Reprinted with permission. Med diet, Mediterranean diet; EVOO, extra-virgin olive oil.

**Table 1 nutrients-13-03269-t001:** Average nutrient composition of tree nuts and oeanuts (per 100 g) [15,16].

Nuts	Energy (kJ)	Protein (g)	Fiber (g)	Fat (g)	SFA (g)	MUFA (g)	PUFA (g)	LA (g)	ALA (g)	Phytosterols (g)
Almonds	2409	21.1	12.5	49.9	3.9	31.5	12.2	12.2	0.00	162
Brazil nuts (dried)	2743	14.3	7.5	66.4	15.1	24.5	24.4	20.5	0.05	72
Cashews	2401	15.3	3	46.4	9.2	27.3	7.8	7.7	0.15	120
Hazelnuts	2669	15.0	9.7	60.8	4.5	45.7	7.9	7.8	0.09	115
Macadamias	2995	7.9	8.0	76	11.9	58.9	1.4	1.3	0.21	119
Peanuts	2372	26	8.5	49.2	6.2	24.4	15.6	15.6	0.00	126
Pecans	2891	9.2	9.6	72.0	6.2	40.8	21.6	20.6	1.00	113
Pine nuts (dried)	2816	13.7	3.7	68.4	4.9	18.8	34.1	33.2	0.16	120
Pistachios	2430	20.6	10.0	47	5.4	25.0	14.0	13.2	0.25	272
Walnuts	2738	15.2	6.7	65.2	6.1	8.9	47.2	38.1	9.08	143

Data for raw nuts, except when specified. ALA, ɑ-linolenic acid; LA, linoleic acid; MUFA, monounsaturated fatty acids; PUFA; polyunsaturated fatty acids; SFA, saturated fatty acids.

**Table 2 nutrients-13-03269-t002:** Average composition of selected micronutrients in tree nuts and peanuts (per 100 g) [15,18].

Nuts	Folate (µg)	Calcium (mg)	Magnesium (mg)	Sodium (mg)	Potassium (mg)	Polyphenols (mg)
Almonds	44	269	270	1	733	287
Brazil nuts	22	160	376	3	659	244
Cashews	69	45	260	16	565	233
Hazelnuts	113	114	163	0	680	671
Macadamias	10	70	118	4	363	126
Peanuts	240	92	168	18	705	406
Pecans	22	70	121	0	410	1284
Pine nuts	34	16	251	2	597	58
Pistachios	49	104	106	6	977	1420
Walnuts	98	98	158	2	441	1579

**Table 3 nutrients-13-03269-t003:** Associations of nut consumption with health outcomes and disease risk factors. Summary of scientific evidence.

Disease/Factor	Association	Level of Evidence
Epidemiologic studies		
Cardiovascular disease	Reduction	++
Coronary heart disease	Reduction	++
Stroke	No change//reduction	+/−
Heart failure	No change/reduction	+/−
Atrial fibrillation	Reduction	+
Hypertension	Reduction	+
Diabetes	No change/reduction	+/−
Cognitive dysfunction	Improvement	+
Depression	No change/reduction	+/−
Cancer	Reduction	++
Obesity	No change/reduction	++
All-cause mortality	Reduction	++
Randomized clinical trials		
Blood lipid profile		
Total cholesterol	Reduction *	++
LDL-cholesterol	Reduction *	++
HDL-cholesterol	No change	++
Triglycerides	Reduction *	++
Insulin sensitivity	No change/increase *	+
Diabetes control	Improvement	+
Blood pressure	No change/reduction *	+/−
Inflammation	No change/reduction *	+
Vascular reactivity	Improvement	+
Body weight	No change/slight reduction *	++
Waist circumference	No change/slight reduction *	++
Metabolic syndrome	Improvement or reversion **	+
Type-2 diabetes incidence	No change **	+
CVD incidence	Reduction **	+
Stroke incidence	Reduction **	+
PAD incidence	Reduction **	+
Cognitive function	Improvement *	+

Abbreviations: +/−, equivocal evidence; +, limited evidence from few studies; ++, evidence from many studies; HDL, high-density lipoprotein; LDL, low-density lipoprotein; PAD peripheral artery disease. * Evidence collected in the PREDIMED trial, among others. ** Evidence collected only in the PREDIMED trial.

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
