# Peer review of "Nuts: Natural Pleiotropic Nutraceuticals"

_nutrients, 2021, doi:10.3390/nu13093269_

Round 1

Reviewer 1 Report

In this manuscript, Ros et al. review the health benefits of nut consumption. The manuscript is well written and covers the relevant publications in this field.

Minor concerns

To the best of my knowledge, the term “pleiotropic” is more commonly used in genetics. Please ensure that the term is suitable also for the purpose used in the manuscript.

“Calorie” is a unit of energy. As is the case with other units, they should be used with respective numerical values, as in “60 s”, “100 g”, and “2000 kcal”. When talking about the actual phenomenon, I would recommend using the name of the phenomenon, as in “time”, “weight”, and “energy”. Therefore, I recommend changing “calorie” -> “energy”, in the abstract and on page 2.

On page 2, row 48. You state here that nut consumption has increased DUE TO a number of factors. Do you have any research data to show the cause and effect? If so, please provide references. If not, I would suggest to rather say that the increase has taken place “in parallel with”, or something along those lines.

Page 3, row 119. “…their high-energy and…” -> “…their high energy content and…”

Page 3, lines 123 and 125. Please specify the percentages shown here. Are they percentages of total energy content (E%) or does this refer to a percentage of weight?

Page 5, row 159. “…endothelium-derived nitric oxide (NO) synthesis…” -> “…the synthesis of endothelium-derived nitric oxide (NO)…” This suggestion is made as the sentence continues with a reference to the nitric oxide (“…the main endogenous regulator of vascular tone…”). The sentence reads better if the “NO“ is the last word prior to this latter part of the sentence.

Page 5, line 163. What is meant by “standard serving”?

Table 2, Header. “Average composition of tree nuts and peanuts in selected micronutrients” -> “Average composition of selected micronutrients in tree nuts and peanuts”.

Table 2. Please add a missing bracket to the unit of folate.

Page 6, row 179. “…their fatty fraction…” -> “…their fat fraction…”

Page 6, row 183. “…lower blood cholesterol…” -> “…lower blood cholesterol concentration…”

Page 6, row 184. “in gram doses”, please ensure this is a proper term to use.

Page 7, row 227. “…are the leading cause of death…” -> “…are the leading causes of death…”

Page 8, row 249 and Page 10, row 344. Please consider not using a term “diabetic” when describing “individuals with diabetes”. The latter is typically the preferred term.

Page 9, row 290. HF has not been abbreviated. And as this abbreviation has not been used in the remaining text, I would suggest to write it out (instead of using abbreviation).

Page 10, row 335. “…was not entered as a covariate in the adjustment model…” -> “…was excluded from the model…”

Page 10, row 344. “…analysis of diabetic participants…” You never actually “analyze people”, you analyze data collected from these people. Please rephrase.

Page 10, row 354. Please remove “the” from the sentence: “…unique the nutrient composition…”

Page 11, rows 411 and 421. I believe the disease is most typically written as “Alzheimer’s disease”.

Page 17, row 662. “…with carbohydrate foods…” -> “… with carbohydrate containing foods…” OR “carbohydrate-rich”.

Page 22, row 909. AFIB has not been abbreviated. And as this abbreviation has not been used in the remaining text, I would suggest to write it out (instead of using abbreviation).

Page 23, row 941. Please remove one of the “must be”s.

Page 23, row 952. “…lower LDL-cholesterol…” -> “…lower LDL-cholesterol concentration…”

Page 23, row 953 and Table 3. “Epidemiologic studies…” -> “Epidemiological studies…”

Author Response

Responses to Reviewer 1

Comments and Suggestions for Authors

In this manuscript, Ros et al. review the health benefits of nut consumption. The manuscript is well written and covers the relevant publications in this field.

Thank you for your overall positive comment.

Minor concerns

To the best of my knowledge, the term “pleiotropic” is more commonly used in genetics. Please ensure that the term is suitable also for the purpose used in the manuscript.

“Pleiotropic” can be used to describe multifaceted effects of either genes or drugs. In either use it refers to multiple important actions, usually beneficial, beyond what was originally recognized. For instance, statins are said to have pleiotropic effects (anti-inflammatory, immunomodulatory, anti-atherosclerotic, etc.) beyond their main effect of “cholesterol lowering”. In this review we describe multiple biological effects exerted by nuts, also beyond cholesterol lowering, their best recognized effect. Thus, we believe it is appropriate to use the term in this context, though it is somewhat novel in reference to a food (which performs as a drug, hence a nutraceutical). Even so, all readers will understand its implications as used in the title of the review. 

“Calorie” is a unit of energy. As is the case with other units, they should be used with respective numerical values, as in “60 s”, “100 g”, and “2000 kcal”. When talking about the actual phenomenon, I would recommend using the name of the phenomenon, as in “time”, “weight”, and “energy”. Therefore, I recommend changing “calorie” -> “energy”, in the abstract and on page 2.

Done as suggested.

On page 2, row 48. You state here that nut consumption has increased DUE TO a number of factors. Do you have any research data to show the cause and effect? If so, please provide references. If not, I would suggest to rather say that the increase has taken place “in parallel with”, or something along those lines.

“In parallel with” used as suggested.

Page 3, row 119. “…their high-energy and…” -> “…their high energy content and…”

OK, “content” added.

Page 3, lines 123 and 125. Please specify the percentages shown here. Are they percentages of total energy content (E%) or does this refer to a percentage of weight?

Percentage of weight, as added to text.

Page 5, row 159. “…endothelium-derived nitric oxide (NO) synthesis…” -> “…the synthesis of endothelium-derived nitric oxide (NO)…” This suggestion is made as the sentence continues with a reference to the nitric oxide (“…the main endogenous regulator of vascular tone…”). The sentence reads better if the “NO“ is the last word prior to this latter part of the sentence.

Done, thank you for this useful suggestion

Page 5, line 163. What is meant by “standard serving”?

In the US, a standard serving of nuts is 1-oz (slightly more than 28 g). To round up the number, it is also customary to consider 30 g a serving (as used in the PREDIMED trial).

Table 2, Header. “Average composition of tree nuts and peanuts in selected micronutrients” -> “Average composition of selected micronutrients in tree nuts and peanuts”.

Header of Table 2 changed as suggested.

Table 2. Please add a missing bracket to the unit of folate.

Done

Page 6, row 179. “…their fatty fraction…” -> “…their fat fraction…”

Fatty replaced by fat.

Page 6, row 183. “…lower blood cholesterol…” -> “…lower blood cholesterol concentration…”

Done.

Page 6, row 184. “in gram doses”, please ensure this is a proper term to use.

Doses of 1 gram or higher.

Page 7, row 227. “…are the leading cause of death…” -> “…are the leading causes of death…”

Done.

Page 8, row 249 and Page 10, row 344. Please consider not using a term “diabetic” when describing “individuals with diabetes”. The latter is typically the preferred term.

Changed in the two places.

Page 9, row 290. HF has not been abbreviated. And as this abbreviation has not been used in the remaining text, I would suggest to write it out (instead of using abbreviation).

HF omitted as abbreviation.

Page 10, row 335. “…was not entered as a covariate in the adjustment model…” -> “…was excluded from the model…”

Done as suggested.

Page 10, row 344. “…analysis of diabetic participants…” You never actually “analyze people”, you analyze data collected from these people. Please rephrase.

Rephrased as suggested.

Page 10, row 354. Please remove “the” from the sentence: “…unique the nutrient composition…”

Done.

Page 11, rows 411 and 421. I believe the disease is most typically written as “Alzheimer’s disease”.

Changed.

Page 17, row 662. “…with carbohydrate foods…” -> “… with carbohydrate containing foods…” OR “carbohydrate-rich”.

Changed.

Page 22, row 909. AFIB has not been abbreviated. And as this abbreviation has not been used in the remaining text, I would suggest to write it out (instead of using abbreviation).

AFIB written out.

Page 23, row 941. Please remove one of the “must be”s.

Done.

Page 23, row 952. “…lower LDL-cholesterol…” -> “…lower LDL-cholesterol concentration…”

Done.

Page 23, row 953 and Table 3. “Epidemiologic studies…” -> “Epidemiological studies…”

Changed.

Reviewer 2 Report

  • The review is well written and everything is comprehensible.
  • The number of the papers used is considerable for such a review.
  • The present review is of great significance since authors have gathered information about nut consumption and its correlation with lower cancer incidence and cancer mortality, favorable effects on cognitive function and depression and so on; thus it will be helpful for the readers interested in several different fields.

Author Response

Responses to Reviewer 2

Comments and Suggestions for Authors

  • The review is well written and everything is comprehensible.
  • The number of the papers used is considerable for such a review.
  • The present review is of great significance since authors have gathered information about nut consumption and its correlation with lower cancer incidence and cancer mortality, favorable effects on cognitive function and depression and so on; thus it will be helpful for the readers interested in several different fields.

Thank you for your praise of the manuscript.